## Research Article

mental health; anxiety; depression; satisfaction with life; caregivers; psychiatric illness; Malaysia; family stress model

**Corresponding author:**
Norhayati Ibrahim;
Email: yatieibra@ukm.edu.my

# Factors associated with depression, anxiety, and satisfaction with life among Malaysian parental caregivers of adolescent psychiatric patients: A cross-sectional study

Choy Qing Cham[1] ⬥, Norhayati Ibrahim[1,2], Clarisse Roswini Kalaman[1] ⬥, Meng Chuan Ho[3], Uma Visvalingam[4], Farah Ahmad Shahabuddin[5], Fairuz Nazri Abd Rahman[6], Mohd Radzi Tarmizi A. Halim[7], Manveen Kaur Harbajan Singh[8], Fatin Liyana Azhar[8], Amira Najiha Yahya[9], Samsilah Roslan[10] and Ching Sin Siau[11]

[1]Center for Healthy Ageing and Wellness (H-CARE), Faculty of Health Sciences, Universiti Kebangsaan Malaysia, Kuala Lumpur, Malaysia; [2]Institute of Islam Hadhari, Universiti Kebangsaan Malaysia, Bangi, Malaysia; [3]Centre for Pre-U Studies, UCSI University (Springhill Campus), Port Dickson, Malaysia; [4]Department of Psychiatry and Mental Health, Hospital Sungai Buloh, Sungai Buloh, Malaysia; [5]Department of Psychiatry, Hospital Bahagia Ulu Kinta, Tanjung Rambutan, Malaysia; [6]Psychiatry Department, Faculty of Medicine, Universiti Kebangsaan Malaysia, Kuala Lumpur, Malaysia; [7]Faculty of Business, Economics and Social Development, Universiti Malaysia Terengganu, Kuala Nerus, Malaysia; [8]Department of Psychological Medicine, Faculty of Medicine, University Malaya, Kuala Lumpur, Malaysia; [9]Department of Educational Psychology and Counselling, Faculty of Education, Universiti Malaya, Kuala Lumpur, Malaysia; [10]Department of Foundation of Education, Faculty of Educational Studies, Universiti Putra Malaysia, Serdang, Malaysia and [11]Center for Community Health Studies (ReaCH), Faculty of Health Sciences, Universiti Kebangsaan Malaysia, Kuala Lumpur, Malaysia

## Abstract

Depression and anxiety are prevalent mental health issues worldwide, especially among parental caregivers. By expanding the family stress model, this cross-sectional study investigated the relevant factors associated with depressive symptoms, anxiety symptoms, and satisfaction with life among Malaysian parental caregivers of adolescent psychiatric patients. Data were collected through questionnaires ($N = 207$) across five major public hospitals through convenience sampling. Participants answered questionnaires measuring financial strain, caregiver burden, relationship quality, belief in mental illness, perceived COVID-19 stress, satisfaction with health services, depressive symptoms, anxiety symptoms and life satisfaction. Findings revealed that relationship quality among spouses, COVID-19 stress and caregiver burden were significantly correlated with anxiety symptoms, depressive symptoms and satisfaction with life. The multiple regression model also suggested that depressive symptoms ($\beta = .613$, $p < .001$), anxiety symptoms ($\beta = .657$, $p < .001$) and relationship quality among spouses ($\beta = .264$, $p < .001$) were the most influential predictors of anxiety symptoms, depressive symptoms and satisfaction with life respectively. By addressing the mental health needs of parental caregivers, this study can contribute to improving the overall quality of care and support provided to adolescent patients and their caregivers in Malaysia and beyond.

## Impact statement

Recognising the mental health challenges faced by parental caregivers in Malaysia can inform global efforts to develop and enhance caregiver support programs. By providing targeted interventions, resources, and services to address caregiver depression, anxiety, and low life satisfaction, countries worldwide can promote caregiver well-being and resilience. This can have a positive ripple effect on the mental health of caregivers in diverse cultural and healthcare contexts. The findings on the level of mental health among Malaysian parental caregivers of adolescent psychiatric patients can inform the development and implementation of global mental health policies. By recognising the impact of caregiver mental health on patient outcomes and overall family dynamics, policymakers can prioritise mental health support for caregivers within healthcare systems. This can influence policy changes, resource allocation, and the integration of caregiver mental health in mental healthcare frameworks worldwide.

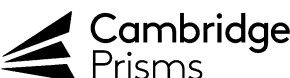



## Introduction

### Introduction to global mental health

Globally, the prevalence of mental and behavioural disorders is a substantial concern that affects approximately 450 million individuals (Walke et al., 2018). In 2015, the global population witnessed a substantial burden, with 4.4% suffering from depressive disorders and 3.6% from anxiety disorders (World Health Organization, 2017). Notably, depressive and anxiety disorders were the main contributors to the worldwide health-related burden of mental illnesses (Santomauro et al., 2021). Among these, depression holds the highest prevalence of 4.3% and is among the largest single cause of mental illness worldwide (Friedrich, 2017). Transitioning to the Malaysian context, the country reports a 2.3% prevalence of depression among adults, as per the 2019 National Health and Morbidity Survey (NHMS; Institute for Public Health, 2020).

### Mental health among caregivers and adolescents in Malaysia

The concern about mental health extends to caregivers, where a local study reported high prevalence rates of depression (32.8%) and anxiety (27.8%) among caregivers (Ivan et al., 2019). In 2015, Malaysia recorded a total of 445,335 outpatient psychiatric cases, marking a significant increase of 16.4% compared to 2014 (Malaysian Healthcare Performance Unit, 2017). Even among children, the gravity of mental health issues is evident, with the 2019 National Health and Morbidity Survey citing a prevalence of 7.9% (Institute for Public Health, 2020). Despite the severity of these mental health challenges, specialised professional services for adolescents are inadequate in Malaysia. For example, the country only has 24 child and adolescent psychiatrists to serve a population of over 30 million, falling substantially short of the WHO's recommended ratio of 1:10,000 (Ng et al., 2018).

### Caregivers and the burden of care

As mental health challenges grow, there is a shift in the provision of care for individuals with mental disorders. It has been expected that the provision of care for 50–90% of people with mental disorders has now been transferred to the obligations of family members after the deinstitutionalization movement (Tamizi et al., 2020). However, this transition places caregivers, who care for family members with psychiatric illnesses, at potential risk for significant burdens and a subsequent decline in their own health status (Souza et al., 2017). Adding to these difficulties, the COVID-19 pandemic and the resulting lockdown measures have presented extensive challenges to parents of psychiatric patients. Caregivers encountered unique burdens during the COVID-19 pandemic (Budnick et al., 2021). Caregivers of psychiatric patients had a tough time ensuring continuous care due to disruptions in mental health services and limited in-person treatments. They had to adapt to remote healthcare, handle medications, and ensure safety, all amid increased uncertainty, making the caregiving burden heavier for both caregivers and patients (Picardi et al., 2021).

### Caregiver roles and their challenges

The term "caregiver" encompasses individuals responsible for patient care at home, executing specific instructions provided by their treating healthcare personnel (Zubaidi et al., 2020). They often provide physical, emotional and financial support for those who cannot take care of themselves due to disabilities, illness or injury (Sherman, 2019). Caregivers are important as they need to take responsibility for most of the needs of care recipients (Aman et al., 2020). However, they are at a high risk of developing mental health disorders (Fernando et al., 2022). Not only that, caregivers' employment, educational prospects, finances and social life can also be affected (Aman et al., 2020). Caregiving often requires a significant amount of time and effort. Caregivers may need to attend to the needs of their loved ones during working hours, leading to frequent absences or disruptions at work (Bijnsdorp et al., 2022). Additionally, caregivers may need to postpone their educational goals or drop out of educational programmes due to the demands of caregiving. This can hinder their personal and professional growth. Caregiving often comes with additional costs, such as medical bills, medications, assistive devices, and home modifications. These expenses can strain caregivers' finances (Chen et al., 2019). Caregivers may experience social isolation as they prioritise caregiving responsibilities over social interactions. This isolation can lead to feelings of loneliness and depression (Ploeg et al., 2020). A systematic review conducted by Cham et al. (2022) revealed that nearly one-third of caregivers of individuals with mental illness, excluding major neurocognitive disorders, suffered from caregiver burden. Research has consistently demonstrated that providing informal family care is a demanding job with detrimental effects on the mental and physical health of caregivers (Del-Pino-Casado et al., 2021).

### Parental caregivers and their crucial role

In the context of parental caregivers, their role assumes increasing significance in the treatment outcomes of psychiatric patients (Ran et al., 2016). Parental caregivers assume diverse roles, most of the time without adequate training, including the identification of mental health issues, advocacy, provision and management of mental health services for the patient. Parental caregivers are often the first to notice changes in their child's behaviour, emotions, or well-being. They observe shifts in mood, habits, or social interactions that may signal the presence of mental health issues (Di Renzo et al., 2020). Caregivers often take on the role of advocates for their children within the healthcare system. They schedule appointments, communicate with healthcare providers, and ensure their child receives appropriate assessments and treatment (Brock-Baca et al., 2023). Additionally, caregivers play a key role in coordinating and ensuring their child's access to mental health services. This includes arranging therapy sessions, medication management, and other necessary interventions (Matsuzawa et al., 2020). Caregivers monitor their child's progress in therapy or treatment, regularly assessing whether interventions are effective and they provide feedback to healthcare professionals to adjust treatment plans as needed (Maunder and McNicholas, 2021).

Literature indicates that caregivers of mental health patients suffer from high caregiver burdens and psychological distress (Ong et al., 2016). Some risks that predispose to a higher caregiver burden are the female gender of the caregiver, unemployment, being a parent, and having health problems (Ran et al., 2016). In addition, marital conflict could arise from the stresses faced by husband and wife.

### The family stress model

The family stress model (FSM) framework (Masarik and Conger (2017) theorises that a family under stress will experience issues in the interaction between children and parents, leading to a child's difficulty in adjusting. In the original FSM, it is postulated that

economic hardship and economic pressure lead to parental psychological distress. Parental psychological distress leads to disrupted parenting via interparental relationship problems. This, in turn, leads to a child's adjustment problems. As suggested by Masarik and Conger (2017), various environmental stressors could also contribute to parental distress, other than economic hardship. In this study, we attempted to expand the original FSM to take into account families that deal with mental illness in one of the dependent family members. Based on literature reviews, we found that three more factors could be added to the original FSM, namely the caregiving factor (operationalized as caregiver burden), the cultural factor (stigma), and the healthcare system factor (service satisfaction). As mentioned in the literature review above, caregiving burden may place stress on the family system as parental caregivers are faced with multiple roles in the family and towards the patient (Aman et al., 2020; Brock-Baca et al., 2023). Stigma, as a cultural factor, could also burden the family system, with parental caregivers reporting being isolated or judged by their community members (Vadivelan et al., 2020). Service satisfaction, as a healthcare system factor, could influence the quality of care provided to the patients. There have been allegations of inadequate facilities, incompetent staff, excessive wait times, and negative attitudes in public hospitals (Ting et al., 2019; Garney et al., 2021). Inadequate healthcare services may increase the burden of care, which may have a detrimental effect on the patient's results. Along with the economic factor (monthly household income and perceived financial stress) and the interparental relationship factor (relationship quality), all these factors are hypothesised to be significantly associated with parents' mental health issues (higher depressive and anxiety symptoms) and lower satisfaction with life. In turn, poor parental mental health could lead to a particular parenting style that affects the adolescent patient's treatment adherence and treatment outcomes.

There were many studies conducted on the mental health of caregivers of psychiatric patients in Western countries (Rexhaj et al., 2017; Souza et al., 2017; Ribé et al., 2018; Schuster et al., 2020), but limited research was done in Malaysia. Thus, we proposed that the risk factors (relationship quality, financial status, stigma, service satisfaction, caregiver burden and COVID-19 stress) will be associated with the parental caregiver's mental health, in the aspects of anxiety symptoms, depressive symptoms and satisfaction with life.

## Methodology

### Study location

This cross-sectional study was conducted in five major hospitals under the Ministry of Health (MOH) Malaysia between December 2021 and September 2022, which are Hospital Bahagia Ulu Kinta, Hospital Tuanku Ampuan Najihah, University Malaya Medical Centre, Hospital Canselor Tuanku Muhriz, and Hospital Putrajaya.

### Participants

This study is part of a larger study examining the association between parental caregivers and child well-being. Data on 220 parent–child dyads were collected. In this study, we have extracted data on parental caregivers to focus on investigating factors associated with parental depressive symptoms, anxiety symptoms, and life satisfaction.

The sample size was calculated using G*Power analysis, with an effect size of 0.15, $\alpha$ at 0.05, power at 90%, and 17 predictors in the input parameters, which were used to determine the study sample size (Hair et al., 2010). With the addition of 15% of the participant's dropout to the formula, a total of 201 parental caregivers were targeted for recruitment through the convenience sampling method. The inclusion criteria included (1) being a Malaysian citizen, and (2) being able to provide informed consent and assent. Excluded participants were those whose children had schizophrenia spectrum disorders or active psychosis. This decision was made before data collection for the main study involving parental-child dyads, considering children with active psychosis may not be able to answer the questionnaire meaningfully. Thus, by default, their parental caregiver was excluded from the study as well.

The researchers used universal sampling to identify the participants after receiving the psychiatrist's list of potential participants (Martinez-Mesa et al., 2016). The data of 13 participants was removed due to incomplete questionnaires or missing data. In the end, data analysis was conducted on 207 participants, and it fulfilled the targeted sample size.

### Measures

The following questionnaires were self-administered to the parental caregivers:

### Demographic information

Demographic information was collected on participants, which included age, gender, race, religion, marital status, monthly income, occupation status and relationship to the patient.

### Relationship quality index

The three-item abbreviated version of the relationship quality index (RQI; Norton, 1983) was used to measure relationship quality. This version was previously employed in another study (McGill et al., 2016). The three items were "We have a good relationship", "My relationship with my partner/spouse is strong" and "My relationship with my partner/spouse makes me happy". Participants answered on a seven-point Likert scale ranging from 1 = "very strongly disagree" to 7 = "very strongly agree". A higher total score indicated a better relationship quality. This questionnaire was back-translated into the Malay language in a study involving the general public in Malaysia and recorded an internal consistency reliability of $\alpha = .93$ (Siau et al., 2020). The internal consistency reliability of the scale score in this study was $\alpha = .93$.

### Caregiver burden

The Zarit burden interview (ZBI) screening version is a four-item questionnaire employed to measure the level of caregiver burden (Zarit et al., 1985). Participants answered on a five-point Likert scale ranging from 0 ("Not at all") to 4 ("Nearly always"). The Malay ZBI demonstrated excellent internal reliability ($\alpha = .898$) among parental caregivers of cancer patients (Shim et al., 2017). The internal consistency reliability of the scale score in this study was $\alpha = .83$.

### Financial strain

Subjective financial strain was measured based on three questions: (1) "In general, would you say that you/your family has more money than you need, just enough for your needs or not enough to meet your needs?", (2) "How difficult is it for you/your family to pay monthly bills–very difficult, somewhat, not very, or not at all difficult?", and if the family was experiencing difficulty, they were

asked (3) "Did this difficulty start in the past 12 months?" Families were considered to be experiencing financial strain if they did not have enough money, had difficulty paying monthly bills, and had problems in the past 12 months (O'Campo et al., 2004). This scale was back-translated into the Malay language by the study researchers and independent language experts. A consensus meeting was held among the researchers to harmonise these versions and ensure comprehensibility.

### Mental health stigma

Two domains from the belief towards mental illness (BMI) questionnaire which were the five-item dangerousness and 10-item social skills domains, were chosen for this study (Hirai and Clum, 2000). The incurability domain was excluded due to the decision of the researchers to shorten the questionnaire, which was a recommendation of the ethics board, in order to prevent fatigue to the participants. Participants responded on a six-point Likert scale ranging from 1 ("Completely disagree") to 6 ("Completely agree"). This scale was back-translated into the Malay language and recorded an overall internal consistency of $\alpha = 0.80$ in a study involving secondary school students (Ibrahim et al., 2019). The BMI has good internal consistency reliability with Cronbach's $\alpha = .89$ in this study. The internal consistency reliability of the dangerousness and social skills domains in this study were $\alpha = .78$ and $\alpha = .84$, respectively.

### Parent satisfaction of healthcare services

The Parent Satisfaction Scale (PSS) is an 11-item questionnaire that measures parents' satisfaction with health services provided for their children (Almeida et al., 2015). Participants answered on a five-point Likert scale ranging from 1 ("Strongly disagree") to 5 ("Strongly agree"). The Malay-PSS demonstrated an acceptable reliability coefficient (Raykov's $\rho = .851$) when tested among parents of children with autism (Adib et al., 2018). The internal consistency reliability of the scale score in this study was $\alpha = .95$.

### COVID-19 stress

The perceived stress scale modified for COVID-19 (PSS-10-C) was used to assess how participants perceived their stress levels during the COVID-19 pandemic. This questionnaire consists of 10 self-reported items. This scale featured six items focused on gauging perceived distress and 4 items aimed at evaluating perceived coping. Participants rated their answers on a five-point Likert scale ranging from 0 ("Never") to 4 ("Very often"). The validated Malay version of the scale was used (Ibrahim et al., 2023). According to Ibrahim et al. (2023), this scale displayed good internal consistency reliability, with a Cronbach's $\alpha$ coefficient of. 86, when administered to a group of young adults in Malaysia. The scale obtained internal consistency reliability within the acceptable range in this study for parental caregivers (Cronbach's $\alpha = .68$).

### Depressive symptoms

The Patient Health Questionnaire – 9 (PHQ-9) is a nine-item scale that measures depression in primary care (Kroenke et al., 2001). Participants answer on a four-point Likert scale ranging from 0 ("Not at all") to 3 ("Nearly every day'). The Malay version of PHQ-9 was found to have good internal reliability and validity (Sherina et al., 2012a). A lower score obtained indicates a lower level of depression. The internal consistency reliability of the original scale was $\alpha = .89$, and the internal consistency obtained in this study was $\alpha = .84$.

### Anxiety symptoms

The Generalized Anxiety Questionnaire – 7 (GAD-7) is a seven-item questionnaire used to measure anxiety in primary care (Spitzer et al., 2006). Participants answer on a four-point Likert scale ranging from 0 ("Not at all") to 3 ("Nearly every day"). The Malay version of the GAD-7 was found to have good internal reliability (Cronbach's $\alpha = .74$; Sherina et al., 2012b). A lower score indicates a lower level of anxiety. The internal consistency reliability of the original scale and the internal consistency obtained in this study were both $\alpha = .92$.

### Satisfaction with life

The Satisfaction with Life Scale is a five-item scale aimed at measuring participants' life satisfaction (Diener et al., 1985). The Malay SWLS has good internal consistency (Cronbach's $\alpha = 0.83$) when validated among a community sample (Swami and Chamorro-Premuzic, 2009). Participants answered on a five-point Likert scale ranging from 1 ("Strongly disagree") to 5("Strongly agree"). A higher score obtained indicated higher life satisfaction. The internal consistency reliability of the original scale was $\alpha = .87$, and the internal consistency obtained in this study was $\alpha = .89$.

### Statistical analysis

All statistical analyses in this study were carried out using IBM SPSS Statistics for Windows, Version 27.0 (IBM Corp., Armonk, NY, USA). A descriptive analysis was conducted to interpret the demographic variables. Independent variables that were statistically significant in the bivariate analysis ($p < .05$) were included in the multiple regression analysis. The categorical variable "Difficulty in paying monthly bills in the past 12 months" was dummy-coded, and the "No" category served as the reference group for anxiety symptoms. For depressive symptoms, "Diagnosed with mental illness", "Buddhists", "Christians", "Hindu" and "Others" were dummy coded, and the "No" and "Islam" categories for all the religions served as the reference group respectively. The categorical variables "Family financial situation", "Difficulty in paying monthly bills" and "Difficulty in paying monthly bills in the past 12 months" were dummy coded, and the "Enough", "Not Difficult" and "No" categories served as the reference groups for satisfaction with life respectively. The predictors of caregivers' mental health in this study were analysed using multiple linear regression analysis. Prior to the multiple regression analyses, data normality was tested based on an absolute $z$-value for skewness and kurtosis of <3.29 (Kim, 2013). Multicollinearity was examined by assessing Variance Inflation Factor (VIF) values, which needed to be below 10, and tolerances, which should be above 0.1, for the predictor variables. The presence of multivariate outliers was identified using the Maximum Mahalanobis Distance, which exceeded the critical chi-square ($\chi^2$) value for degrees of freedom equal to the number of predictor variables ($k$) (Allen et al., 2014). After the analyses, it was found that the data distribution met the normality assumption for each variable. No multivariate outlier or collinearity was found in the final model.

## Results

### Demographic profile

Table 1 shows the demographic profile of psychiatric patient caregivers. There were 115 (55.6%) female participants, while males comprised 92 participants, equivalent to 44.4%. The predominant racial group among the participants was Malay, comprising 78.3%

**Table 1.** Sociodemographic of the caregiver of adolescents with mental illness

| Variable | n | % | Anxiety symptoms | | | Depressive symptoms | | | Satisfaction with life | | |
|---|---|---|---|---|---|---|---|---|---|---|---|
| | | | Mean (SD) | t/F statistic | p-value | Mean (SD) | t/F statistic | p-value | Mean (SD) | t/F statistic | p-value |
| Gender | | | | 1.302 | .195 | | .652 | .515 | | −.540 | .590 |
| Male | 92 | 44.4 | 4.07 (4.842) | | | 6.41 (5.370) | | | 17.37 (4.392) | | |
| Female | 115 | 55.6 | 3.29 (3.433) | | | 5.97 (4.511) | | | 17.69 (4.049) | | |
| Race | | | | 3.571 | .015* | | 3.331 | .021* | | 1.767 | .155 |
| Malay | 162 | 78.3 | 3.17 (3.777) | | | 5.71 (4.817) | | | 17.85 (4.340) | | |
| Chinese | 26 | 12.6 | 4.88 (4.761) | | | 7.38 (4.665) | | | 16.92 (2.770) | | |
| Indian | 16 | 7.7 | 6.13 (5.414) | | | 9.19 (5.319) | | | 15.50 (4.502) | | |
| Others | 3 | 1.4 | 4.33 (3.786) | | | 4.00 (3.000) | | | 17.33 (2.082) | | |
| Religion | | | | 3.421 | .010* | | 2.640 | .035* | | 1.002 | .408 |
| Islam | 166 | 80.2 | 3.22 (3.769) | | | 5.70 (4.787) | | | 17.83 (4.299) | | |
| Buddha | 13 | 6.3 | 6.31 (5.345) | | | 7.38 (4.682) | | | 16.54 (3.688) | | |
| Hindu | 10 | 4.8 | 4.80 (6.321) | | | 10.10 (5.216) | | | 16.70 (4.296) | | |
| Christian | 16 | 7.7 | 3.81 (3.987) | | | 7.63 (5.239) | | | 16.31 (3.135) | | |
| Others | 2 | 1.0 | 3.50 (0.707) | | | 5.00 (2.828) | | | 15.00 (5.657) | | |
| Marital status | | | | .740 | .460 | | −.706 | .481 | | .048 | .961 |
| With a spouse | 189 | 91.3 | 3.70 (4.198) | | | 6.09 (4.880) | | | 17.55 (4.213) | | |
| Single | 18 | 8.7 | 2.94 (3.298) | | | 6.94 (5.230) | | | 17.50 (4.148) | | |
| Highest qualification | | | | 2.641 | .051 | | 1.419 | .238 | | 2.387 | .070 |
| Primary/secondary school | 66 | 31.9 | 2.55 (3.149) | | | 5.20 (4.236) | | | 18.23 (4.209) | | |
| Diploma | 53 | 25.6 | 4.51 (5.213) | | | 6.96 (5.177) | | | 16.32 (4.367) | | |
| Bachelor's degree | 75 | 36.2 | 4.03 (4.064) | | | 6.45 (5.416) | | | 17.64 (3.951) | | |
| Master's degree/ doctor of philosophy | 13 | 6.3 | 3.31 (2.840) | | | 6.15 (3.184) | | | 18.54 (4.156) | | |
| Diagnosed with mental illness | | | | −.646 | .519 | | −2.214 | .028* | | −1.041 | .312 |
| Yes | 20 | 9.7 | 4.2 (3.636) | | | 8.45 (4.796) | | | 18.45 (4.199) | | |
| No | 187 | 90.3 | 3.57 (4.180) | | | 5.92 (4.865) | | | 17.45 (4.197) | | |
| Family financial situation | | | | −1.068 | .287 | | −.455 | .649 | | 2.071 | .040* |
| Enough | 153 | | 3.45 (4.131) | | | 6.07 (4.995) | | | 17.9 (4.148) | | |
| Not enough | 54 | | 4.15 (4.109) | | | 6.43 (4.673) | | | 16.54 (4.210) | | |
| Difficulty in paying monthly bills | | | | −1.577 | .118 | | .237 | .813 | | 2.312 | .023* |
| Not difficult | 149 | | 3.34 (3.952) | | | 6.21 (4.926) | | | 17.99 (.983) | | |
| Difficult | 58 | | 4.4 (4.491) | | | 6.03 (4.888) | | | 16.41 (4.546) | | |
| Difficulty in paying monthly bills in the past 12 months | | | | −2.258 | .026* | | −.521 | .603 | | 2.969 | .003** |
| Yes | 66 | | 4.61 (4.378) | | | 6.42 (4.826) | | | 16.3 (4.382) | | |
| No | 141 | | 3.19 (3.938) | | | 6.04 (4.952) | | | 18.13 (3.993) | | |

*p < 0.05,
**p < 0.01,
***p < 0.001.

of the total with 162 participants. Chinese individuals comprised 12.6% with a total of 26 participants, while Indian participants constituted 7.7% with 16 individuals. Additionally, there were three participants (1.4%) from other racial backgrounds. In terms of religion, the majority of participants identified as Muslim, with 80.2%, totaling 166 individuals. A total of 16 participants (7.7%) were Christians, followed by 13 participants (6.3%) being Buddhists, 10 participants (4.8%) being Hindu participants and two participants (1%) were from other religious backgrounds. A total of 189 participants (91.3%) were married and 18 participants (8.7%) were separated or divorced from their spouses. In terms of educational attainment among the participants, the majority, comprising 36.2%, held a bachelor's degree (75 participants). Following closely were 31.9% of participants (66 individuals) who had completed only primary or secondary school. Those with a diploma accounted for 25.6%, totalling 53 participants, while a smaller proportion, 6.3%, consisted of individuals with master's or doctorate degrees, amounting to 13 participants. Furthermore, 187 participants (90.3%) were never diagnosed with a mental illness, while 20 participants (9.7%) had previously been diagnosed with a mental illness. One-way ANOVA and independent *t*-test analyses showed that there was a significant difference in terms of race ($F$ (3, 203) = 3.571, $p$ = .015) and religion ($F$ (4, 202) = 3.421, $p$ = .010) in caregiver anxiety symptoms. A significant difference in race ($F$ (3, 203) = 3.331, $p$ = .021), religion ($F$ (4, 202) = 2.640, $p$ = .035) and having a mental disorder diagnosis ($t$ (205) = -2.214, $p$ = .028) in caregiver depressive symptoms were also identified. High multicollinearity was presented between race, religion and depression. Therefore, race was excluded from the final depression regression model.

Pearson's correlations shown in Table 2 suggested that relationship quality among spouses ($r$ = −.142, $p$ = .041), COVID-19 stress ($r$ = .424, $p$ < .001) and caregiver burden ($r$ = .223, $p$ = .001) were significantly correlated with anxiety symptoms of the caregivers. Relationship quality among spouses ($r$ = −.203, $p$ = .003), COVID-19 stress ($r$ = .281, $p$ < .001), caregiver burden ($r$ = .290, $p$ < .001) and anxiety symptoms ($r$ = .687, $p$ < .001) were significantly correlated with depressive symptoms of the caregivers. While for the satisfaction of life of the caregivers, results showed a significant correlation with social skill ($r$ = .184, $p$ = .008), relationship quality ($r$ = .310, $p$ < .001), COVID-19 stress ($r$ = −.241, $p$ < .001), caregiver burden ($r$ = −.211, $p$ = .002), service satisfaction ($r$ = −.175, $p$ = .012), anxiety symptoms ($r$ = −.323, $p$ < .001) and depressive symptoms ($r$ = −.259, $p$ < .001).

### Factors associated with caregivers' anxiety symptoms, depressive symptoms and satisfaction with life

Variables with significant results in the bivariate analyses were included in multiple linear regression analyses with caregivers' anxiety symptoms, depressive symptoms and satisfaction with life as the dependent variables.

Concerning anxiety symptoms, multiple linear regression was carried out to determine the influence of relationship quality, caregiver burden, COVID-19 stress, difficulty in paying monthly bills in the past 12 months, satisfaction with life and depressive symptoms (Table 3). This was a statistically significant model ($F$ (6, 200) = 40.649, $p$ < .001). The model showed that a 53.6 % variance in anxiety symptoms was accounted for by the predictors. The analysis suggested that depressive symptoms ($\beta$ = .613, $p$ < .001) were the most influential predictor in the model and relationship quality

($\beta$ = .041, $p$ = .412) of the participant was the least influential predictor in the model.

While for depressive symptoms, the model was statistically significant ($F$ (6, 200) = 34.962, $p$ < .001), explaining 35.0 % of the variance in depressive symptoms (Table 4). The analysis suggested that anxiety symptoms ($\beta$ = .657, $p$ < .001) were the most influential predictor in the model and satisfaction with life ($\beta$ = −.006, $p$ < .913) of the participant was the least influential predictor in the model. Diagnosed with mental illness ($\beta$ = .104, $p$ = .044), caregiver burden ($\beta$ = .127, $p$ = .022) and anxiety symptoms ($\beta$ = .657, $p$ < .001) were shown to be statistically significant predictors of depressive symptoms.

For satisfaction with life, the model was statistically significant ($F$ (10, 196) = 6.704, $p$ < .001) with 21.7% of the variance on financial status, COVID-19 stress, relationship quality, caregiver burden, social skills, parents' service satisfaction, depressive symptoms and anxiety symptoms of the participants (Table 5). The analysis suggested that relationship quality ($\beta$ = .264, $p$ < .001) was the most influential predictor in the model and difficulty in paying monthly bills ($\beta$ = .003, $p$ = .983) of the participant was the least influential predictor in the model. Relationship quality ($\beta$ = .264, $p$ < .001), social skills ($\beta$ = .150, $p$ = .017) and anxiety symptoms ($\beta$ = −.229, $p$ = .013) were shown to be statistically significant predictors of satisfaction with life. Financial status, COVID-19 stress, caregiver burden, parents' service satisfaction and depressive symptoms of the participants were shown not to be statistically significant predictors of satisfaction with life.

## Discussion

Based on the FSM, this study aimed to study the association between relationship quality, financial status, stigma, service satisfaction, caregiver burden, and COVID-19 stress with depressive symptoms, anxiety symptoms, and satisfaction with the life of Malaysian parental caregivers of adolescent psychiatric patients. As hypothesized, relationship quality, stigma, service satisfaction, caregiver burden, and COVID-19 stress were significantly associated with the mental health of parental caregivers of adolescent psychiatric patients.

### Factors associated with caregivers' anxiety symptoms

The findings revealed that caregivers who have had difficulty paying monthly bills in the past 12 months and have depressive symptoms are more likely to suffer from anxiety symptoms. This finding is supported by Hu et al. (2018), who mentioned that the poor financial situation of caregivers was one of the factors that affected the caregiver's emotions. The financial strain resulting from the failure to meet monthly financial commitments amplifies feelings of helplessness and hopelessness, contributing to anxiety symptoms. Simultaneously, the demanding nature of caregiving for psychiatric patients, coupled with the financial burden, leads to heightened anxiety as caregivers worry about the future, their capacity to sustain care, and the well-being of their loved ones (Rahman et al., 2018).

In Malaysia, as in many cultures, there can be strong expectations for family members to provide unwavering support to their loved ones (Ahmad et al., 2021), especially during times of hardship. While this cultural value can be optimistic about supporting family unity, it can also establish a stigma around seeking help for mental health issues. Caregivers may feel constrained to keep their

**Table 2.** Mean, standard deviation, internal consistency reliability, and Pearson's correlations of correlates of mental health

| Variable | Mean (SD) | $\alpha$ | Age | Dangerousness | Social skills | Relationship quality | COVID-19 stress | Caregiver burden | Parents' service satisfaction | Anxiety symptoms | Depressive symptoms | Satisfaction with life |
|---|---|---|---|---|---|---|---|---|---|---|---|---|
| Age | 46.47 (6.38) | | 1 | | | | | | | | | |
| Dangerousness | 18.99 (4.42) | .78 | .124 | 1 | | | | | | | | |
| Social skills | 41.00 (7.82) | .84 | .058 | .628*** | 1 | | | | | | | |
| Relationship quality | 16.38 (4.18) | .97 | .010 | −.065 | .027 | 1 | | | | | | |
| COVID-19 stress | 15.77 (5.20) | .68 | .075 | −.030 | −.082 | −.103 | 1 | | | | | |
| Caregiver burden | 5.78 (3.15) | .83 | .024 | −.087 | .008 | −.076 | .293*** | 1 | | | | |
| Parents' service satisfaction | 17.34 (5.42) | .95 | .002 | −.045 | −.131 | −.005 | .094 | .141* | 1 | | | |
| Anxiety symptoms | 3.63 (4.13) | .92 | .046 | .050 | −.032 | −.142* | .424*** | .223** | .056 | 1 | | |
| Depressive symptoms | 6.16 (4.90) | .84 | .078 | .029 | −.076 | −.203** | .281*** | .290*** | .119 | .687*** | 1 | |
| Satisfaction with life | 17.55 (4.20) | .89 | −.089 | .056 | .184** | .310*** | −.241*** | −.211** | −.175* | −.323*** | −.259*** | 1 |

*$p < .05$,
**$p < .01$,
***$p < .001$.

**Table 3.** Multiple linear regression of factors associated with anxiety symptoms

| Variable | B | 95% CI Lower | 95% CI Upper | Beta | p-value |
|---|---|---|---|---|---|
| (Constant) | −.808 | −3.548 | 1.932 | | .561 |
| Relationship quality | .041 | −.057 | .139 | .041 | .412 |
| Caregiver burden | −.066 | −.199 | .066 | −.050 | .326 |
| COVID-19 stress | .180 | .098 | .262 | .226*** | <.001 |
| Difficulty in paying monthly bills in the past 12 months (no)[1] | | | | | |
| Difficulty in paying monthly bills in the past 12 months (yes) | .618 | −.245 | 1.480 | .070 | .159 |
| Satisfaction with life | −.117 | −.220 | −.015 | −.119* | .025 |
| Depressive symptoms | .516 | .429 | .602 | .613*** | <.001 |

[1]Reference group.
*p <. 05,
**p <. 01,
***p <. 001.

**Table 4.** Multiple linear regression of factors associated with depressive symptoms

| Variable | B | 95% CI Lower | 95% CI Upper | $\beta$ | p-value |
|---|---|---|---|---|---|
| (Constant) | 4.598 | 1.266 | 7.930 | | .007 |
| Diagnosed with mental illness (no)[1] | | | | | |
| Diagnosed with mental illness (yes) | 1.717 | .045 | 3.389 | .104* | .044 |
| Relationship quality | −.111 | −.232 | .010 | −.095 | .072 |
| Caregiver burden | .198 | .028 | .367 | .127* | .022 |
| COVID-19 stress | −.044 | −.149 | .060 | −.047 | .404 |
| Satisfaction with life | −.007 | −.136 | .122 | −.006 | .913 |
| Anxiety symptoms | .780 | .645 | .915 | .657*** | <.001 |
| Islam[1] | | | | | |
| Buddhist | −1.200 | −3.242 | .842 | −.060 | .248 |
| Hindu | 1.424 | −.876 | 3.725 | −.062 | .223 |
| Christian | 1.173 | −.651 | 2.997 | .064 | .206 |
| Others (religion) | −1.347 | −6.237 | 3.543 | −.027 | .588 |

[1]Reference group.
*p <. 05,
**p <. 01,
***p <. 001.

struggles hidden, fearing that discussing their mental health concerns openly would be perceived as a sign of weakness or inadequate caregiving (Ibrahim et al., 2019).

The co-occurrence of difficulties in paying monthly bills and depressive symptoms among caregivers of psychiatric patients in Malaysia carries significant implications. Financial strain could worsen mental health issues, which can lead to an increase in anxiety symptoms. The potential for a decreasing impact on the caregiver's general well-being is what makes this important. Financial hardships might hinder both the caregiver and the psychiatric patient's access to vital services and healthcare, potentially lowering the standard of care (Munsell et al., 2016). This complex interaction highlights the crucial need for targeted support mechanisms, including financial assistance programs, accessible mental health services, and awareness campaigns to reduce stigma and encourage

caregivers to seek help (Monnapula-Mazabane and Petersen, 2023). Avoiding these issues could affect a cycle of distress for both caregivers and psychiatric patients, eventually impacting the broader mental health landscape in Malaysia.

### Factors associated with caregivers' depressive symptoms

Caregivers who suffer from burden or have anxiety symptoms will be more likely to suffer from depressive symptoms. These findings are in line with Del-Pino-Casado et al. (2019), which underscore the significant role of caregiver burden as a pivotal risk factor in the development of depressive symptoms among informal caregivers.

In terms of race comparison, Indians were more likely to have depressive symptoms. Although the findings are not generalizable, the results of this study were associated with past studies (Cheah

**Table 5.** Multiple linear regression of factors associated with satisfaction of life

| Variable | B | 95% CI | | $\beta$ | p-value |
|---|---|---|---|---|---|
| | | Lower | Upper | | |
| (Constant) | 13.825 | 9.401 | 18.249 | | <.001 |
| Family financial situation (enough)[1] | | | | | |
| Family financial situation (not enough) | .189 | −1.362 | 1.740 | .020 | .811 |
| Difficulty in paying monthly bills (not difficult)[1] | | | | | |
| Difficulty in paying monthly bills (difficult) | .024 | −2.111 | 2.158 | .003 | .983 |
| Difficulty in paying monthly bills in the past 12 months (no)[1] | | | | | |
| Difficulty in paying monthly bills in the past 12 months (yes) | −1.251 | −3.425 | .923 | −.139 | .258 |
| Relationship quality | .266 | .138 | .393 | .264*** | <.001 |
| Caregiver burden | −.143 | −.318 | .033 | −.107 | .111 |
| Social skills | .081 | .015 | .147 | .150* | .017 |
| Parents' service satisfaction | −.089 | −.186 | .008 | −.115 | .072 |
| COVID-19 stress | −.033 | −.146 | .081 | −.040 | .572 |
| Depressive symptoms | .021 | −.130 | .171 | .024 | .787 |
| Anxiety symptoms | −.233 | −.416 | −.050 | −.229* | .013 |

[1]Reference group.
*p <. 05,
**p <. 01,
***p <. 001.

et al., 2020; Lugova et al., 2021). The high frequency of Indians may be attributed to the poor and economically marginalised status of the Indian minority in Malaysia (Rashid and Tahir, 2015). In the specific context of Malaysia, where traditional family values emphasise the primary responsibility of families in providing care based on principles of mutual support and duty, the caregiving role often imposes emotional and financial strains on caregivers. These strains can potentially exacerbate or even trigger depressive symptoms (Suriawati et al., 2015). Furthermore, these results are supported by empirical evidence suggesting that anxiety symptoms may serve as a precursor to depression (Price et al., 2016). The persistent anxiety experienced by caregivers increases their perception of burden, giving rise to a repeated pattern of emotional distress. This increasing impact of caregiver burden and anxiety together wear down an individual's mental resilience and coping mechanisms, ultimately contributing to the emergence of depressive symptoms, as supported by the theory that anxiety is indeed a risk factor for depression (Shek et al., 2022).

The significance and implications of the interplay between caregiver burden and anxiety leading to depression among caregivers of psychiatric patients in Malaysia cannot be overstated. The development of depression can have great effects on their quality of life (Wu et al., 2018). This dynamic has the potential to impact the care provided to psychiatric patients, potentially leading to a decline in the patient's mental health due to compromised caregiving. Furthermore, the healthcare system may face increased demands if caregivers themselves need care and support for their mental health issues. Moreover, the societal stigma surrounding mental health in Malaysia may prevent caregivers from seeking help, further exacerbating their depression (Ong et al., 2016). Identifying and addressing this issue is not only crucial for the caregivers themselves but also for the general mental health landscape of Malaysia, highlighting the urgent need for targeted support

services and raising awareness about mental health challenges faced by caregivers.

### Factors associated with caregivers' satisfaction with life

Life satisfaction is a concept that may be used to assess the calibre of healthcare services and gauge how well people are doing (Danacı and Koç, 2018). The findings in this study revealed that caregivers who have negative belief that psychiatric patients have poor social skills, good relationship quality with their spouse, a low burden and a low level of anxiety symptoms are more likely to have greater satisfaction with life. The findings were aligned with Grevenstein et al. (2019), who stated that the positive association between elevated life satisfaction, is rooted in enhanced relationship quality and refined social skills.

In Malaysia, where cultural values emphasise the significance of family ties, this suggests that caregivers' increased life satisfaction can have a ripple effect on caregiving relationships (Tan et al., 2019). This, in turn, can contribute to more robust support for individuals with mental health conditions, aligning with the deeply ingrained cultural context. Moreover, Rippon et al.'s (2020) emphasis on the perceived quality of relationships as a protective factor reinforces the connection between relationship quality and heightened life satisfaction among caregivers. Conversely, the diminishment of life happiness, as indicated by Jiang et al. (2020), can amplify the weight of caregiving responsibilities.

It is worth noting that several studies, including those by Bhatia (2020), Yu et al. (2020), and Rajendran et al. (2022), consistently underscore the adverse association between anxiety and life satisfaction. Within the Malaysian caregiving context, this underscores the profound impact of anxiety on caregivers. Anxious caregivers tend to report diminished life satisfaction, highlighting the imperative need for comprehensive support mechanisms addressing

caregivers' mental health and overall life satisfaction to ensure the well-being of both caregivers and those they care for (Diener and Seligman, 2004).

In the Malaysian context, where caregiving plays a pivotal role in supporting individuals with psychiatric illnesses, the significant implications of caregivers who exhibit good relationship quality with their spouse and report low anxiety symptoms experiencing greater life satisfaction cannot be understated. Firstly, these caregivers are likely to provide more effective and compassionate care, as they are better equipped to manage the emotional challenges inherent in caregiving (Giesbrecht et al., 2012). Secondly, their own well-being is likely to be positively impacted, lowering the risk of caregiver burnout and mental health problems (Pinquart and Sorensen, 2004). Moreover, in a culture that highly values familial bonds, this improved life satisfaction among caregivers can improve the overall support system for individuals with psychiatric illnesses, contributing to improved outcomes and quality of life for both caregivers and care recipients (Ibrahim et al., 2018). Understanding these implications underscores the need for tailored support and interventions that promote caregivers' mental health, relationship well-being, and overall life satisfaction in the Malaysian healthcare system.

## Limitations and future recommendations

The current study is subject to several limitations. Due to the cross-sectional nature of this study, it is only possible to draw general conclusions regarding the magnitude and direction of correlations between the components that were examined. However, the questionnaire solely used self-report scales for measurement, which are subjective and sensitive to biases such as social desirability, participant mood, or the urge for self-improvement. Only internal consistency reliability was reported for the Malay-translated relationship quality index, financial strain, and scale since validity indices were not available in the original studies that had translated them. Therefore, we could not rule out issues such as redundancy and reduced sensitivity of items. In addition, the Incurability domain of the belief in mental illness scale was excluded from this study due to the need to shorten the questionnaire, resulting in leaving out an important aspect of stigma. To fully comprehend the problems of caregivers' mental health in Malaysia, more research is required. To better comprehend the factors influencing mental health and its impact on caregivers, longitudinal cohort studies should be conducted.

## Conclusions

In conclusion, caregiver attention for persons with mental illness should thus increase given their significance in the treatment outcomes of psychiatric patients in the age of deinstitutionalization. Recognising the formidable mental health challenges faced by parental caregivers in Malaysia not only holds significance within the nation but also has far-reaching implications for global efforts to support caregivers. By acknowledging and addressing the factors influencing depression, anxiety, and low life satisfaction among caregivers, countries around the world can learn from Malaysia's experiences and develop targeted interventions and support programs. These initiatives can bolster caregiver well-being and resilience, ultimately benefiting the mental health of caregivers in diverse cultural and healthcare settings. Importantly, the insights gained from studying Malaysian parental caregivers of adolescent psychiatric patients should serve as a catalyst for the development of comprehensive global mental health policies regarding this vulnerable and overburdened group. This holistic approach not only improves the lives of caregivers but also enhances patient outcomes and strengthens family dynamics, ultimately promoting a more compassionate and inclusive healthcare landscape for families dealing with psychiatric illnesses.

**Open peer review.** To view the open peer review materials for this article, please visit http://doi.org/10.1017/gmh.2024.5.

**Data availability statement.** The data that support the findings of this study are available from the corresponding author (N.I.) upon reasonable request.

**Acknowledgements.** We are very grateful to all participants who contributed to this study. We would like to thank the Director General of Health Malaysia for his permission to publish this article.

**Author contribution.** Conceptualization: N.I., A.N.Y., C.S.S.; Formal analysis and data interpretation: C.R.K., C.S.S.; Methodology: C.Q.C., N.I., C.R.K., C.S.S.; Project administration: M.C.H., U.V., F.A.S., F.N.A.R., M.R.T.A.H., M.K., and F.L.A.; Writing–original draft: all authors; Writing–review and editing: all authors. All authors have read and approved the final version of the manuscript.

**Financial support.** This research received its funding from the Fundamental Research Grant Scheme (FRGS/1/2020/SS0/UCSI/02/1) from the Ministry of Higher Education, Malaysia.

**Competing interest.** The authors declare none.

**Ethical statement.** Ethical approval for this study was obtained from the Medical Research and Ethics Committee (MREC), Ministry of Health Malaysia (NMRR-20-754-53871) and UKM Research Ethics Committee (JEP-2021-886).

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
