## [Reviewer Report]

Editor-in-Chief

Cambridge Prisms: Global Mental Health

August 3, 2023

Dear Editor:

I am pleased to submit an original research article titled “Factors associated with depression, anxiety, and satisfaction with life among Malaysian parental caregivers of adolescent psychiatric patients: A cross-sectional study” for consideration for publication in Cambridge Prisms: Global Mental Health.

Recognizing the mental health challenges faced by parental caregivers in Malaysia can inform global efforts to develop and enhance caregiver support programs. By providing targeted interventions, resources, and services to address caregiver depression, anxiety, and low life satisfaction, countries worldwide can promote caregiver well-being and resilience. This can have a positive ripple effect on the mental health of caregivers in diverse cultural and healthcare contexts. The findings on the level of mental health among Malaysian parental caregivers of adolescent psychiatric patients can inform the development and implementation of global mental health policies. By recognizing the impact of caregiver mental health on patient outcomes and overall family dynamics, policymakers can prioritize mental health support for caregivers within healthcare systems. This can influence policy changes, resource allocation, and the integration of caregiver mental health in mental healthcare frameworks worldwide.

This manuscript has not been published and is not under consideration for publication elsewhere. We have no conflicts of interest to disclose. All the respective authors had read and approved this manuscript’s overall content before submitting it for publication. We sincerely hope that the written manuscript is appropriate for your journal and will be given full consideration for review.

Thank you for your consideration!

Yours faithfully,

Norhayati Ibrahim.

Senior Lecturer

Center for Healthy Ageing and Wellness (H-CARE),

Faculty of Health Sciences, 

Universiti Kebangsaan Malaysia 

Jalan Raja Muda Abdul Aziz,

Kuala Lumpur 50300, Malaysia.

yatieibra@ukm.edu.my

---

## [Reviewer Report]

Comments to the Author

The strength of this paper is that it reports original research to expand the Family Stress Model. This research specifically aims to understand factors associated with depression symptoms, anxiety symptoms, and satisfaction with life among Malaysian parental caregivers of adolescent psychiatric patients, as well as understand the impact on care. Thank you for undertaking this important work.

ABSTRACT

Lines 38-44 – ‘expanding the family stress model…………..Malaysian parental caregivers of adolescent psychiatric patients’ - This sentence seems incomplete; rephrase to provide clarity.

PAGE 4

Line 6 - Add “by’ before 16.4%.

Line 8 – Provide a percentage or rate instead of ‘424,000’ children which is easier to interpret and place in context of the population.

Line 45 – Describe how the caregivers’ employment, educational prospects, finances and social life are affected.

PAGE 5

Family-Healthcare Stress Model – It is not clear if the authors of this study are expanding the model as part of this work. If that is the case, then provide a detailed process undertaken to expand the model.

Lines 11- 15 –‘ Parental caregivers assume diverse roles……’

Are the caregivers trained to provide the roles described here? Provide details of the training process or experience required.

PAGE 7

Line 11 -Careful consideration needs to be given to language use. In particular, it is no longer accepted practice to refer to ‘target’ population. Using ‘priority’ is preferred.

Line 13 – ‘Around’ 220 participants…….. – Provide the exact sample size.

PAGE 8

Line 13 – ‘Financial Strain’ - Describe how the survey instruments were developed, whether the questions were standardized or study-specific. How was comprehensibility established and culturally verified?

INSTRUMENTS

COVID-19 Stress is mentioned throughout the paper, however there is no indication on how this construct was measured. Describe how the survey instruments were developed, whether the questions were standardized or study specific. How was comprehensibility established and culturally verified?

PAGE 10

Line 20 – ‘Phase 1’ of the study is referred here for the for the first time. It is not clear what consist of phase one and how many phases are included in the study. This needs to be described in detail in the methods section.

PAGE 11

Line 49 – How was COVID-19 stress measured?

PAGE 13

Line 8 – Change ‘relationship’ to association.

Line 19 – Based on Tables 3 and 4 COVID-19 stress was not significantly associated with mental health.

---

## [Reviewer Report]

Abstract

In the abstract perhaps the authors wanted to say the study focused on factors. The sentence seemed incomplete. The method in the abstract should be a bit more elaborative to have a concise understanding, for example, the sampling method and the questionnaires used.

Introduction

A bit disorganized and lacks coherence. The authors talk about the prevalence of depression and then suddenly move to talk about the rates of depression and anxiety among caregivers. The mental health repercussions of caregivers are presented in a scattered manner. There should be a paragraph with all types of mental health consequences. Introductions should present a coherent story about the topic under study. I would recommend reorganizing the introduction.

It has been expected that the provision of care for 50–90% of people with mental disorders has now been transferred to the obligation of family members after the deinstitutionalization movement (Sharif et al. 2020). This has been claimed by Tamizi et al. (2020) not Sharif et al. 2020. Citing the original study seems more accurate.

Mentioning studies focusing on the intervention irrespective of the type (i.e., dementia or psychosis) is not the point – rather the point should be the scarcity of studies documenting mental health problems among caregivers.

In the introduction section- the authors claim that little research exists on the mental health of caregivers of people with psychiatric illnesses in Malaysia. In the participants section- they excluded caregivers with children diagnosed with psychosis. This is counterintuitive.

The authors used COVID-19 stress as a variable but the introduction does not offer any context of COVID-19. What is the relevance? How has the burden been experienced by the caregivers during the pandemic? This would justify the use.

Measures

Did the author use the Malaysian version of the Relationship Quality Index (RQI)? This should be specified. If not, the use of this version should be justified. Besides, the psychometric properties of the measure need to be mentioned. While the higher Cronbach’s alpha for this study might seem good, it might also indicate potential issues suggesting biases such as redundancy of items (items are clearly measuring similar construct(s) and reduced sensitivity (items are not capturing variability in responses) or social desirability bias.

The reason for using two domains (Dangerousness and Social Skills) and excluding Incurability needs to be justified. The idea of incurability is one of the strong concepts related to mental health stigma. Please mention the number of items in each domain (Dangerousness and Social Skills). Was the scale validated in the Malaysian language?

Statistical Analysis

Assumptions for normality need to be clarified. Regression analysis is sensitive to multicollinearity (and the way multicollinearity was investigated) and outliers (the ways outliers were identified). Reporting of these issues is recommended.

Since the independent variables are categorical- it is strongly suggested to use one variable as a reference in the multiple regression.

Discussion

The discussion is the weak section of the manuscript. The results and discussions do not provide any novel insight. The factors affecting anxiety and depression are ¬quite understandable and one can tell it from (clinical) experience and general sense. The authors should discuss the significance of these factors such as why they are related to anxiety and depression and what are the potential implications of these factors on their quality of providing care. The underlying mechanism between these factors and anxiety as well as depression would add novelty to this work. Without this—this study provides no additional insights other than an understanding of the Malaysian context.

This section also lacks a discussion of why differences in racial identity and religion are related to anxiety and depression.

The significance and potential implication of the variables found associated with anxiety, depression, and satisfaction with life should be described in a detailed manner with Malaysia as an example.

---

## [Reviewer Report]

The authors adequately addressed the comments. Therefore, I would recommend accepting the manuscript. However, there are a few issues that might be worth looking at.

The Family Stress Model (FSM): What is FMS? Is it a typo or a shorter form of family stress? This should be clarified.

The majority of the participants were Muslims. Islam is a religion and participants can never be a religion. Consider changing it.

It is absolutely necessary to describe the numbers in the demographic profiles so readers can take a look at the table and find the figures.

The authors need to adhere to either American or British English for the entire manuscript.

---

## [Reviewer Report]

I wanted to extend my sincere appreciation for your diligent efforts in addressing the previous comments provided on your manuscript.

I am pleased to acknowledge the significant contribution your paper would make in expanding the Family Stress Model through original research. Your study, which focuses on understanding the factors associated with depression symptoms, anxiety symptoms, and satisfaction with life among Malaysian parental caregivers of adolescent psychiatric patients, is indeed a commendable endeavor. Thank you for undertaking this important work, and I look forward to seeing the positive impact it will undoubtedly have in the field.

---

## [Reviewer Report]

Thank you for the responses to the comments you sent.

Please may you address the minor comments from one of the reviewers.

---

## [Reviewer Report]

The authors have adequately addressed the comments. Therefore, I would recommend accepting the manuscript. Congratulations to the authors!